# PRIPRO: A Comparison of Classification Algorithms for Managing Receiving Notifications in Smart Environments

**João Antônio Martins [1,*], Iago Sestrem Ochôa [1,2], Luis Augusto Silva [1,*], André Sales Mendes [3], Gabriel Villarrubia González [3] and Juan De Paz Santana [3] and Valderi Reis Quietinho Leithardt [1,4,5,*]**

[1] Laboratory of Embedded and Distributed Systems-LEDS, University of Vale do Itajaí, Itajaí SC88302-901, Brazil; iago.ochoa@edu.univali.br

[2] Departamento de Informática e Redes de Computadores, Instituto Federal Catarinense (IFC), Brusque 88354-300, Brazil

[3] Expert Systems and Applications Lab, Faculty of Science, University of Salamanca, Plaza de los Caídos s/n, 37008 Salamanca, Spain; andremendes@usal.es (A.S.M.); gvg@usal.es (G.V.G.); fcofds@usal.es (J.D.P.S.)

[4] Departamento de Informática, Universidade da Beira Interior, 6200-001 Covilhã, Portugal

[5] COPELABS, Universidade Lusófona de Humanidades e Tecnologias, 1749-024 Lisboa, Portugal

* Correspondence: joao_martins@edu.univali.br (J.A.M.); luis.silva@edu.univali.br (L.A.S.); valderi@univali.br (V.R.Q.L.)

**Abstract:** With the evolution of technology over the years, it has become possible to develop intelligent environments based on the concept of the Internet of Things, distributed systems, and machine learning. Such environments are infused with various solutions to solve user demands from services. One of these solutions is the Ubiquitous Privacy (UBIPRI) middleware, whose central concept is to maintain privacy in smart environments and to receive notifications as one of its services. However, this service is freely performed, disregarding the privacy that the environment employs. Another consideration is that, based on the researched related work, it was possible to identify that the authors do not use statistical hypothesis tests in their solutions developed in the presented context. This work proposes an architecture for notification management in smart environments, composed by a notification manager named Privacy Notification Manager (PRINM) to assign it to UBIPRI and to aim to perform experiments between classification algorithms to delimit which one is most feasible to implement in the PRINM decision-making mechanism. The experiments showed that the J48 algorithm obtained the best results compared to the other algorithms tested and compared.

**Keywords:** smart environments; notification management; machine learning

---

## 1. Introduction

Technology is increasingly being incorporated into people's daily lives, becoming more distributed and no longer traditional across various areas of its activities, establishing a new concept, contextualized as the Internet of Things (IoT) [1]. Since many of the objects (electronic components, communication sensors, and mobile devices) that surround people's daily lives are connected, a large amount of information will be generated due to data collection and transmission.

Ordinary everyday places can become intelligent environments when they respond to the presence of people in a versatile manner, meeting their specific needs with the help of IoT objects embedded in the environment [2]. Consequently, people do not notice that they use a computer system directly but understand that the physical environment interfaces with the interaction of the computer system

embedded there. Such environments are part of a distributed system that is a set of software running on one or more computers and coordinating actions by messaging [3].

One of the technologies that assist IoT devices in intelligent environments, and commonly used in distributed systems, is middleware (MW), which is a resource manager that offers your applications the ability to share and deploy these resources efficiently in a network [4]. In addition to resource management, MW offers services similar to those found in an operating system, such as application communication facilities, security services, accounting services, and failure recovery.

In work developed by [5], an MW system was proposed for privacy control and management in intelligent environments called Ubiquitous Privacy (UBIPRI). The central concept is to enable devices to meet the needs of users or environments as a whole, while adapting to different environments and their infrastructure, and adapting device limitations, and environmental privacy. Another goal of UBIPRI is to classify the type of user access in smart environments based on variables such as profile type, user frequency, environment type, and day of the week. Thus, providing users with the availability of services present in the environment accessed and specific actions of these services. One of the MW layer assignments is presented in Figure 1, and it is divided into modules, with each module having a specific task and role in the system.

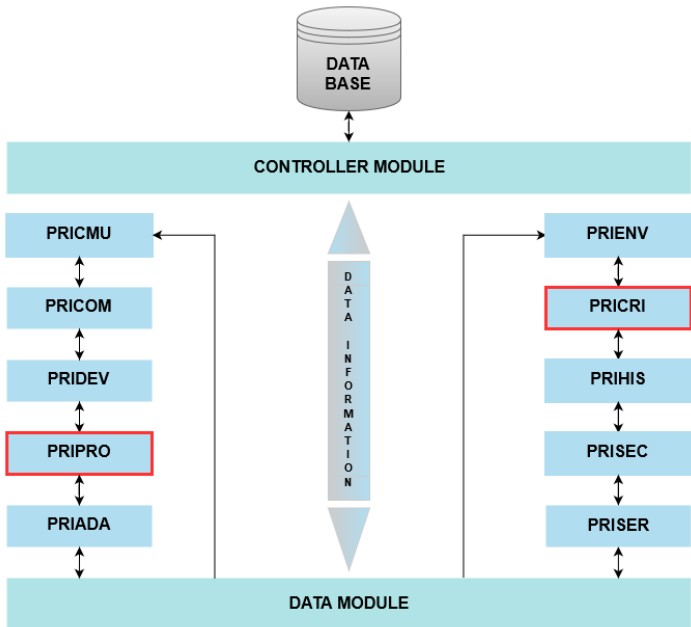

**Figure 1.** Privacy manager model. Adapted from UBIPRI (2015) [6].

According to Figure 1, Privacy Information Management and Control Module User (PRICMU), Privacy Communications (PRICOM), and Privacy Device (PRIDEV) modules are responsible for the management, privacy control, communication and devices, respectively. The Privacy Adapt (PRIADA) module is responsible for adaptation management and control. Privacy Environment (PRIENV) is the environment-related attributes registration module. Privacy History (PRIHIS) is the module for storing and processing information related to user history. Privacy Security (PRISEC) is the module related to user safety and the environment. Privacy Services (PRISER) is the environment service management module. Connecting to all modules, the Data Module processes variables and parameters received from other modules. The Controller Module is the module that receives access requests and performs the control of the database directly in the tables.

The modules described above are not relevant to the development of the work, in contrast, the modules Privacy Profiles (PRIPRO) and Privacy Criteria (PRICRI) are of high relevance. The PRIPRO module is responsible for performing control transactions that are related to user profile management, aiming to distribute and direct synthesized information to the next modules. This information is

adapted appropriately according to the individual privacy of the user and their profile adhered to by UBIPRI. The PRICRI module has rules, criteria, and environment definitions such as access, use, sharing, location, and other variables that can be added, changed, or modified pointing out that each environment has unique characteristics, such that their definitions are treated individually by the other modules that have specific controls.

One of the services that UBIPRI provides is the receipt of notifications for IoT devices in smart environments that are performed freely without the intervention of MW, i.e., disregarding the privacy that the environment employs and which users should expect. Therefore, it is noted that the related work in the context of notification management in smart environments do not use statistical hypothesis tests as a complement to statistical evaluation.

From the gap found in UBIPRI that results in the privacy issue of notifications in smart environments, it is necessary to intelligently manage their receipt, as it is an MW of privacy control and management. Therefore, to ensure the privacy that environments employ, we propose the architecture of a notification manager. Having as its main component, a Decision Maker (DM) mechanism intelligently implemented with a Machine Learning (ML) algorithm belonging to the category of classifying supervision-tasks, for managing receiving notifications from users using UBIPRI.

This paper presents a comparison of classification algorithms to determine which one is the most viable to implement in the PRINM notification manager DM mechanism that will be developed in future work on UBIPRI middleware. It also presents the modeling of notifications' management architecture in the context of intelligent environments. Therefore, this paper aims to report the activities of delimitation and use of classification algorithms and statistical hypothesis testing, generation of artificial datasets, tests and comparisons of classification algorithms and application scenario testing.

This article consists of six sections: Section 2 presents information related to managing work notifications in smart environments; Section 3 presents the methods and materials for the development of the experiments; Section 4 refers to the proposed architecture and application scenario; Section 5 describes the methodology of the experiments performed and their results; and finally Section 6 presents the conclusions and contributions obtained.

## 2. Related Work

First, the authors in [7] restricts itself to detecting disruptive phone calls that are a major source of annoyance to users. To this end, they evaluated six types of learning algorithms, namely: SVM (support vector machine), NB (naive Bayes), KNN (K-nearest neighbors), RUSBoost, GP (genetic programming), and AR (association rule learning). The dataset used for the assessment was collected over a period of 16 weeks with the help of a mobile app. Similar to this work, we also used the NB, SVM, and KNN algorithms, and collected data from a mobile application to create the application scenario test.

Following notification-related studies, the authors at [8] developed an architecture of an intelligent notification system that uses classification algorithms to manage the receipt of notifications according to contextual perception and user habits. The system consists of modules that monitor the environment and users, collecting information to send them to a DM mechanism. The primary relationships with this work are the comparison of the classification algorithms on the classification precision metric, the use of artificial dataset, and the system that implements a classification algorithm in the DM mechanism.

The authors at [9] report in their article, a proposal for location verification and user confirmation in smart environments, in the context of notification control and management. User verification and notification control are performed based on parameters such as environment type, user profile type, location, time criteria, priority, and user preferences. The authors' work is also based on one of the modules of UBIPRI, being PRISER. Furthermore, this work is based on the modules PRIPRO and PRICRI.

In the work of [10], a system has been developed to reduce manual user efforts by addressing and receiving relevant notifications by wireless communication in a university setting. In development, the *Knuth–Morris–Pratt* (KMP) algorithm was applied to a real dataset with the following attributes:

admin, dept, notice, notice, read-by, registration details, staff, and user table. Similar to this work, attributes related to the context was used to perform the notification management.

The reference work [11] discusses in their article an assessment of an artificial dataset in a notification management system. In general, the set has the characteristics of the notification content, user context, and the receiving method, together with the synthetically entered data. In the evaluation, the fuzzy inference system (FIS) algorithm was used to verify the behavior of the generated artificial dataset. The primary relationship with this work is the generation and use of an artificial dataset for PRINM evaluation.

Table 1 presents a synthesis of related work, pointing to the use of classification algorithms, artificial datasets, if the proposed comparison of algorithms on the classification precision metric are used, and if statistical hypothesis tests are used. Comparisons of literature with approaches are defined as: (i) yes, literature treats the approach; (ii) no, the literature does not address the approach; (iii) partial, the literature partially addresses the approach.

**Table 1.** Summary of related work.

| Work | Authors | Uses Classification Algorithms | Uses Artificial Data | Uses Algorithm Comparison | Uses Hypothesis Tests Statistics |
|---|---|---|---|---|---|
| [7] | Smith (2014) | Yes | No | No | No |
| [8] | Corno (2015) | Yes | Yes | Yes | No |
| [11] | Fraser (2017) | No | Yes | No | No |
| [10] | Ghodse (2018) | No | No | No | No |
| [12] | Martins (2018) | Yes | Yes | Yes | Partial |
| [9] | Silva (2019) | Yes | No | No | No |
| This Work | This Work | Yes | Yes | Yes | Yes |

The work presented in [12] does not have a direct relationship with the notification management context, but it is a part of a series of research that is related to UBIPRI. When compared to this work on the notification management approach, only the works [10,11] do not use classification algorithms to manage notifications, but they still have a similarity in artificial dataset approaches. Using artificial datasets is not a good practice when it comes to using ML approaches. However, this practice is becoming increasingly utilized, as in the works [8,11]. The works [7,9] do not address the comparison of classification algorithms over the classification accuracy metric, since there is a range of algorithms to be studied and compared. Only the work [8] addresses this comparison.

It is clear to realize that none of the related work use any statistical hypothesis test as a complement to statistical analysis. This consequently and partially affects the solutions developed by the authors, because with the application of a single statistical hypothesis test appropriate to the context of these works, it would be possible to analyze the statistical differences of the algorithms when compared thoroughly. As seen in this paper, the test performed in Section 5.5 identified that three classification algorithms have the same classification performance, even though they obtain their distinct classification precision metric values.

It is worth mentioning that the gap of not using statistical hypothesis tests presented in the related work is applied in the context of notification management in smart environments. Therefore, they were useful for the development and elaboration of experiments that contributed to solving the problem listed in this work. Because of this, this work proposes to perform the activities of delimitation and use of classification algorithms and statistical hypothesis testing, generation of artificial datasets, tests and comparisons of classification algorithms, and application scenario testing. At the end of the work development, it was determined that the J48 algorithm is the most viable for implementation in the PRINM DM mechanism that will be developed in UBIPRI.

## 3. Methods and Materials

We briefly introduce the concept of ML, focusing on the classification task, which presents the learning categories and their respective classification algorithms that are listed and selected. As well

as the study and delimitation of a statistical hypothesis test. Finally, this section includes the process performed to generate artificial datasets.

### 3.1. Machine Learning Concept

Machine learning can be defined as programming that helps computers make decisions using data from examples of past experiences. It is based on a model with parameters to be optimized from learning training data. The model can be predictive, and used to make future predictions, or it can be descriptive for data knowledge [13]. Therefore, there are two aspects of ML for model generation—supervised and unsupervised. Each of them can further be broken down into different types of tasks, such as classification, regression, grouping, and association, which consequently have different characteristics and require different algorithms. For the development of the work, we used the classification task that belongs to supervised learning in the concept of ML. The next subsection will describe and approach this task with greater emphasis.

### 3.1.1. Classification Task

A classification task consists of recognizing models that describe and distinguish classes for the purpose of using the model to predict the class of data that has not yet been classified. The task is also stated as: Given a training dataset along with associated predictive attributes, determine the class attribute for an unassigned test dataset. However, the diversity of classification algorithms that exist to solve various problems is large, so it is necessary to study in order to find out which ones are potentially better applied to certain types of problems [14].

Figure 2 presents the learning categories on which the classification algorithms are based. Each learning category directly affects the computational behavior of the algorithm, delimiting how learning is performed and the generated predictor model.

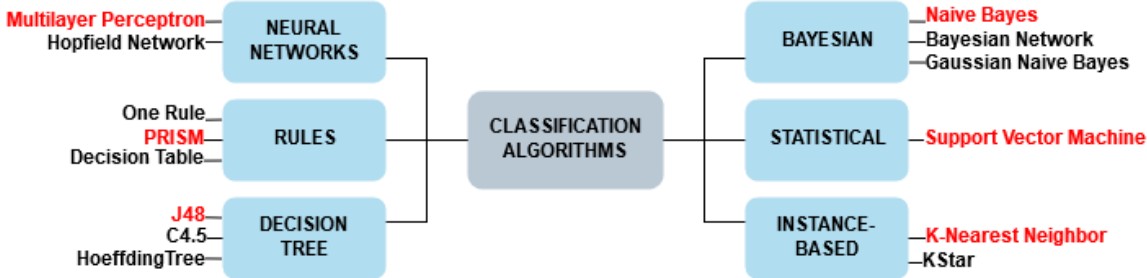

**Figure 2.** Classification task learning categories.

There are several types of algorithms in each learning category, so one algorithm from each category was selected. In the related literature searched, it was not possible to identify classification algorithms that act correctly in the context of notification management in smart environments. Another purpose for this is that classification algorithms act heterogeneously depending on the problem in which it is applied. The selected algorithms were naive Bayes (NB), J48, K-nearest neighbors (KNN), multilayer perceptron (MLP), PRISM, and support vector machine (SVM). The following subsections briefly describe the concept of each classification algorithm.

### 3.1.2. Naive Bayes Algorithm

The NB algorithm is considered to be a simple classifier, and due to its simplicity, has broad applicability in real-time forecasts, news classification, spam filtering, recommendation systems, among others. A peculiarity of classification, called being naive, is that the algorithm disregards the correlation between attributes of a dataset, i.e., it treats each attribute as if it were independent. Because it is a simple algorithm, NB has no adjustable parameters [15].

### 3.1.3. J48 Algorithm

The J48 algorithm is considered to be a fast classifier, and it provides good accuracy rating compared to other classification task algorithms. Derived from its predecessor algorithms ID3, C4.5, and C5.0, J48 builds its tree based on the strategy of division and conquest, by calculating entropy and information gain. A peculiarity of its classification is that the algorithm considers only the most relevant attributes, meaning that, it discards specific attributes that are not relevant to the generation of the predictor model [16]. The main adjustable parameter is the use of pruning, which removes the dirt, thus providing a compact size tree [17].

### 3.1.4. K-Nearest Neighbors Algorithm

The KNN algorithm is the best known and most commonly used instance-based learning algorithm among the scientific community. It is categorized as lazy, as it does not generate a predictor model. Instead, it uses a similarity calculation with all data in the set to classify the new data entered. Therefore, its classification consists of storing training examples, which consequently postpone the processing of training data until new data needs to be classified. The main adjustable parameters are the K variable, which determines the number of nearest neighbors to be discovered, and the similarity calculation to be used [18].

### 3.1.5. Multilayer Perceptron Algorithm

The MLP algorithm consists of a simple system of artificial neurons connected by weights and output signals, which are a function of the sum of inputs for the modified neuron from a linear activation function. The network is divided into three layers: input, hidden, and output. The input layer receives the value vector for network initialization, the hidden layer performs training, and the output layer receives the output vector. The main adjustable parameters are the maximum amount of iterations, learning rate, momentum, and the number of neurons in the hidden layer [19].

### 3.1.6. PRISM Algorithm

The PRISM algorithm is one of the pioneers of its learning category. Others were implemented based on its design. It has in its computational behavior of classification the induction of rules from a dataset. This induction is represented by a fixed set of individual rules for each of the dataset classes. To do so, it has some limitations, such as not generating value enumeration attributes, lacking the robustness of missing values, and performing no pruning. The algorithm has no adjustable parameters [20].

### 3.1.7. Support Vector Machine Algorithm

The SVM algorithm is one of the most efficient classifiers and is used in academia because it can classify data based on mathematical terms. Therefore, it needs a function that describes the factors that must be controlled and guarantees the good performance of the classification. The SVM predictor model generation is based on support vectors, which are used to learn and define the best separation line in the created hyperplane. The algorithm learns the straight line considering the maximum margin defined by it, thus providing the classification between different classes. The main adjustable parameters are kernel and cost [21].

### 3.2. Statistical Hypothesis Tests

The use of statistical hypothesis testing in comparisons of classification algorithms implies an analysis complement between them, indicating whether one algorithm is better than another in a specific task and determining the probability of incorrectly detecting a statistical difference when there is no difference [22]. One of the goals of these tests is to verify the truth of the null hypothesis, which is the statement that there is no distribution difference between samples (datasets). Thus, the hypothesis

verified is H0 (valid and not rejected) or H1 (not valid and rejected) [23]. There are different types of statistical hypothesis testing, namely:

1.  Normality testing that is used to evaluate the assumption of a sample taken from a distributed population [24];
2.  Correlation test that analyzes sample datasets to identify if two variables are related to each other [25];
3.  Association test that reports on the relationship of the statistical association between variables [26];
4.  Variance test comparing the means of different populations [27];
5.  Central tendency test that uses central tendency measures (arithmetic mean, median) to test a probability distribution [28].

From the relationship of the described test types, the central tendency test is the most suitable for the development of the work, because it uses the classification accuracy metric as a measure of central tendency. Table 2 presents the central tendency tests. There are different characteristics among the types of tests presented in Table 2, as follows:

1.  Categorization indicates whether the test is parametric or nonparametric. Parametric tests evaluate the null hypothesis from specific data or parameters (mean, standard deviation, etc.). Nonparametric tests evaluate the null hypothesis from distribution types and group relationships [29];
2.  Variable indicates the types of variables the test supports;
3.  Group, which matches whether the group comparison is individual, paired, or multiple. In this context the classification algorithms are the groups;
4.  Pairing, which corresponds to whether it is paired or unpaired. Paired tests match that the data used for predictor model training are also used to test the predictor model, whereas unpaired tests use one dataset for training and another for testing [30].

Based on the characteristics of the statistical hypothesis tests, we listed those that fit the experiments performed in Section 5, as follows: nonparametric, quantitative, multiple, and paired. The nonparametric characteristic was selected, as it was necessary to identify whether there is really a statistical difference in classification performance between classification algorithms. The classification accuracy metric coincides with the quantitative variable characteristic. Therefore, it is necessary to use multiple comparison tests because the comparison uses six algorithms. Finally, paired tests are best suited, as a single artificial dataset is used for training and testing. Therefore, the Friedman test with these characteristics was listed, to be applied for the comparison of classification algorithms in the Section 5.4 classification precision metric.

By testing the null hypothesis, it is possible to find out if datasets are different from each other, but it is not possible to identify which ones are. Therefore, to solve this impasse, the Friedman test is used, which performs multiple comparisons between equal-sized datasets analyzing the variance and randomization between them. The comparison is made from a ranking presented in Figure 3. To implement, it is necessary to transform raw data into ordered data [31].

In the context of classification algorithms and datasets, $x_{bk}$ represents the placement that the algorithm obtained relative to the dataset in the ranking. This matches that each ranking row corresponds to the random seed value with which the dataset was shuffled and each column corresponds to the algorithm that was applied. Thus, placing $x_{bk}$ corresponds to the value of the classification accuracy metric acquired from the predictor model generated with a given random seed value. Thus, the algorithm with the highest metric value gets the first position in the ranking, the second highest gets the second position, and so on [32]. Equation (1) presents the mathematical calculation of the ranking.

**Table 2.** Types of central tendency tests.

| Name | Categorization | Variable | Group | Pairing |
|---|---|---|---|---|
| Z-test | Parametric | Quantitative | Individual | - |
| T-test | Parametric | Quantitative | Individual | - |
| Wilcoxon for 1 sample | No parametric | Quantitative, ordinal qualitative | Individual | - |
| T-test for 2 samples | Parametric | Quantitative, nominal | Pairs | No paired |
| T-test for 2 samples with different variances | Parametric | Quantitative, nominal | Pairs | No paired |
| T-test paired | Parametric | Quantitative, nominal | Pairs | Paired |
| ANOVA | Parametric | Quantitative, nominal | Multiple | No paired |
| Welch's ANOVA | Parametric | Quantitative, nominal | Multiple | No paired |
| ANOVA for repeated measures | Parametric | Quantitative, nominal | Multiple | Paired |
| Mann-Whitney | No parametric | Quantitative, ordinal qualitative, nominal | Pairs | No paired |
| Wilcoxon Paired | No parametric | Quantitative, ordinal qualitative, nominal | Pairs | Paired |
| Kruskal-Wallis | No parametric | Quantitative, ordinal qualitative, nominal | Multiple | No paired |
| Friedman | No parametric | Quantitative, ordinal qualitative, nominal | Multiple | Paired |
| Test for 1 proportion | Parametric | Nominal | Individual | - |
| Test for 2 proportion | Parametric | Nominal | Pairs | No paired |

**Figure 3.** Friedman test ranking.

From the calculation of Equation (1), the value of the critical difference is the most important, because it indicates whether there is a statistical difference between the summation values of two algorithms in the ranking. This difference is discovered by subtracting these values. Thus, if the result of the subtraction obtained is greater than the critical distance, then it corresponds that the two algorithms are statistically different and that one of them is better in the task adhered to them, that is, in the dataset in which they were applied [33]. Therefore, with the Friedman test, it is possible to identify if there is a statistical difference between classification algorithms in the face of a given dataset when there is such a difference.

$$|R_i - R_j| \geq Z\left(\frac{\alpha}{k(k-1)}\right)\sqrt{\frac{N \times k(k+1)}{6}} \tag{1}$$

where:

- $R_i$ and $R_j$ is the sum of the positions of the algorithms $i$ e $j$ in the ranking;
- $|R_i - R_j|$ is the difference between the sum of the algorithms;
- $Z\left(\frac{\alpha}{k(k-1)}\right)\sqrt{\frac{N \times k(k+1)}{6}}$ is the critical difference.

### 3.3. Artificial Datasets

To perform the experiments, three artificial datasets were generated from a script executed in the NetBeans IDE, consisting of predictor attribute values and classifier attributes arranged in the ARFF file format. All three datasets use the same predictor attributes and different classifier attributes, therefore, for each set a different classification objective is defined as follows:

- Target: classifies which user the notification must be notified to;
- Period: classifies what time of day the notification must be notified;
- Setting: classifies which device configuration notification must be notified.

The number of data instances is precisely the same for each set containing 4320 data. The predictor attributes and their values are shown in Figure 4, and the classifier attributes and their values are shown in Figure 5.

```
@attribute  user  { Member1, Member2, Member3 }
@attribute  profile  { Blocked, Guest, Basic, Advanced, Administrator }
@attribute  environment  { Public, Private, Restrict }
@attribute  activity  { Relevance1, Relevance2, Relevance3 }
@attribute  status  { On, Off }
@attribute  inPeriod  { InMorning, InAfternoon, InNight, InDawn }
@attribute  inTarget  { InMember1, InMember2, InMember3, InAll }
```

**Figure 4.** Predictor attributes.

```
@attribute  outTarget  { OutMember1, OutMember2, OutMember3, OutAll, OutTargetNone }
@attribute  outPeriod  { OutMorning, OutAfternoon, OutNight, OutDawn, OutPeriodNone }
@attribute  outSetting  { OutSilent, OutVibrate, OutCurrent, OutSettingNone }
```

**Figure 5.** Classifier attributes.

The description of the predictive attributes are as follows: (i) user, identifies which user is in the smart environment; (ii) profile, determines the type of user profile used. This attribute is related to the PRIPRO module; (iii) environment, determines which type the environment has. This attribute is related to the PRICRI module; (iv) activity, indicates the relevance of the activity the user is performing in the environment; (v) status, indicates the status of the device that should be notified; (vi) inPeriod, indicates the period that the notification was received in the smart environment; (vii) inTarget, indicates to which user the notification should be notified.

To do so, the classifying attributes are: (i) outTarget, classifies which user the notification must be notified to; (ii) outPeriod, classifies what time of day the notification must be notified; (iii) outSetting, classifies which device configuration notification must be notified;

Predictor and classifier attributes are assigned different types of values and may contain one or more of them that are related to the context of the work. Thus, the user attribute is assigned the values Member1, Member2, Member3, stating that the environment has three members. The profile types are determined with the values Blocked, Guest, Basic, Advanced, Administrator of the attribute profile. There are three distinct types of environment indicated by the Public, Private, Restrict values of the environment attribute. The values attribute the relevance of activities performed by users Relevance1, Relevance2, Relevance3 of the attribute activity, being respectively the first with less, the second with average and the third with high relevance. The On, Off values of the status attribute

indicate respectively whether the device is on or off. The period in which notification was received in the environment is assigned by the InMorning, InAfternoon, InNight, InDawn values of the inPeriod attribute. Finally, the InMember1, InMember2, InMember3, InAll values of the inTarget attribute determine which user the notification should be notified to, either for specific users or for everyone.

Starting with the values of the classifier attributes, the outTarget attribute is assigned the values OutMember1, OutMember2, OutMember3, OutAll, OutNone indicating to which user the notification must be notified, either to specific users or to all. The outPeriod attribute that is assigned the values OutMorning, OutAfternoon, OutNight, OutDawn, OutPeriodNone, indicating the period in which notification must be notified to the user. Finally, the setting that the notification must be notified of is delimited by the OutSilent, OutVibrate, OutCurrent, OutSettingNone values of the outSetting attribute. Values ending with None mean the notification must not be notified.

## 4. Notification Management Architecture

We divided the architecture into three layers which are: the smart environment, UBIPRI, and PRINM. In the smart environment layer, the sensors have the purpose of collecting information about the environmental context in which it operates and the users present in it. The UBIPRI layer receives the information and delivers it to the PRICRI and PRIPRO modules, which then sends the environment and user attributes to PRINM. At the PRINM layer, online services notifications collected by the *receiver* are received, sending attributes and notifications to the DM mechanism. Finally, the DM mechanism classifies for which user, period and device configuration notifications must be notified from the acquired attributes. Figure 6 presents the architecture overview.

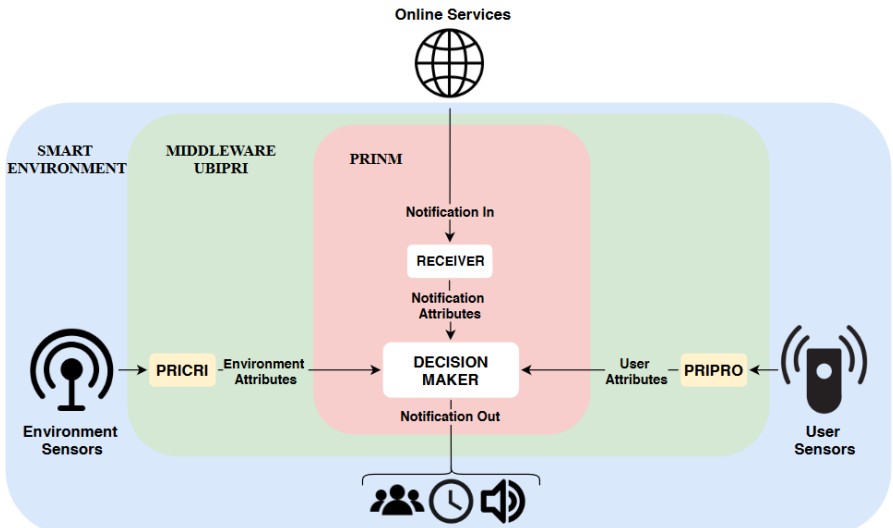

**Figure 6.** Notification management architecture.

Considering the architecture presented in Figure 6 with the theoretical framework of smart environments and the UBIPRI discussed in Section 1, and the basis for the classification algorithms described in Section 3. PRINM is an implementation to be developed in UBIPRI to maintain the privacy of environments in the context of receiving notifications, utilizing an intelligent DM mechanism assigned a classification algorithm that receives attributes regarding the environment, users and notifications. The manager ensures the delivery of notifications according to the privacy managed by UBIPRI in the smart environment in which it operates.

For a better understanding of the proposed architecture, an application scenario was created based on the generated artificial datasets and contextualization presented. The scenario is a car dealership that uses the services of UBIPRI, composed of four different areas, designated showroom, sales, kitchen, and office. Each area has a specific environment type. The scenario also has three users, denominated

customer, employee, and owner, who have their IoT devices attached to UBIPRI. As long as users are in the dealership while receiving notifications, PRINM's decision making will notify them. Figure 7 shows the view of the dealership building and each area with its environment type.

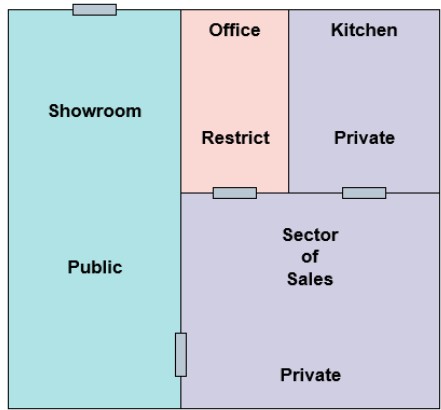

**Figure 7.** Car dealership building.

In order not to increase the scope of PRINM's performance in the application scenario, we determined that it would happen in just one day with pre-established actions for each user. Thus, for a better understanding and contextualization of the scenario, we individually described the actions of each user within the dealership:

- Customer: He arrives at the dealership in the middle of the morning to search for cars to buy in the showroom. He is serviced by the employee, who then directs him to the sales department to negotiate with the owner. He leaves late in the morning and returns midway through the afternoon. He is attended to by the employee in the showroom, who then directs him to the sales department to continue negotiating with the owner. The deal is closed at the owner's office. He leaves late in the afternoon.
- Employee: He arrives at the dealership early in the morning to open it and perform its tasks in the showroom. He takes a break in the kitchen. He serves the customer and forwards it for negotiation with the owner in the sales sector. He takes a break for lunch in the kitchen. He opens the dealership in the afternoon and performs tasks in the showroom. He covers the homeowner on sales tasks and leaves late in the afternoon.
- Owner: He arrives at the dealership, already opened by the employee early in the morning, and performs their tasks in the sales department. He performs tasks in his office and soon after goes to the sales department to attend the customer referred by the employee. He takes a break in the kitchen. In the early afternoon, he performs tasks in the sales department. He meets the customer again, they close the deal in their private office. He takes a break in the early evening in the kitchen. He does some tasks in his office and leaves in the middle of the night.

Considering each user's actions in the developed application scenario, we developed tests in Section 5.6 to evaluate the architecture of PRINM behavior with the most viable classification algorithm delimited in its DM mechanism.

## 5. Experiments and Results

This section presents the experiments performed with the delimited classification algorithms and the artificial datasets generated in Section 3. The tools used were the WEKA GUI, NetBeans, Excel, and RStudio. All tests were performed on a notebook with the following configurations:

- Intel Core i5-5200U;

- CPU: 2.20 GHz;
- RAM: 6.00 GB.

Therefore, we present the test that identifies whether artificial datasets are suitable for use—the test for adjustable parameters, CPU time for training, and classification of the classification algorithms; the comparison test of classification algorithms on the classification accuracy metric; Friedman's test as a complement to statistical analysis; and finally, the application scenario test presenting the applicability of the architecture over PRINM.

### 5.1. Artificial Datasets Test

After the generation of the Target, Period and Setting artificial datasets presented in Section 3.3, it was necessary to perform tests to identify if they are suitable for application in other experiments of this section. For this, we used the rule learning algorithm ZeroR, which aims to generate the baseline of the classification precision metric. This matches that if the metric of the algorithms intended to be used in the dataset is smaller than that of ZeroR, then it is not indicated or appropriate to use these algorithms. Another objective of the algorithm is to predict the majority class of the dataset, that is, it classifies unclassified data with the class that has the most instances in the training dataset [34]. Table 3 presents the tests performed to generate the baseline with the three artificial datasets, using the validation model cross-validation with the value 10.

**Table 3.** ZeroR baseline generator.

| Artificial Dataset | Classification Accuracy | Majority Class |
|---|---|---|
| Target | 80% | OutTargetNone |
| Period | 80% | OutPeriodNone |
| Setting | 75% | OutSettingNone |

Table 3 shows that the baseline in all sets was above 70%, which, according to [35] is considered the appropriate mean for testing in WEKA of the precision metric of classification. The majority class in each dataset references notifications that must not be notified to users in the smart environment, indicating that these values have the most in their datasets. Defining the baseline in each dataset, the algorithms delimited in Section 3.1.1 were applied to identify whether they reach the percent of the classification accuracy metric above the baseline of the ZeroR algorithm in each set of artificial data. Figure 8 presents the tests performed by applying the algorithms to the artificial datasets Target, Period, and Setting, and using the validation model cross-validation with the value 10.

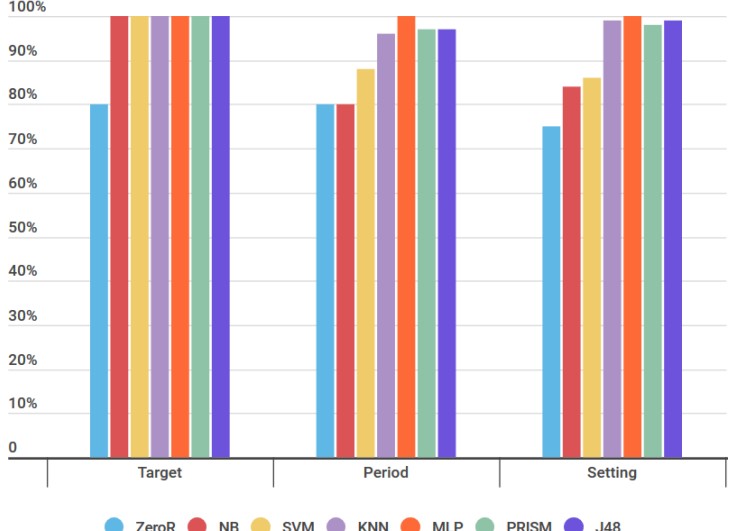

**Figure 8.** Baseline test.

Presenting Figure 8, we noticed that the percentage of the classification accuracy metric of all algorithms in each dataset were above the baseline generated by the ZeroR algorithm in the previous test seen in Table 3. This indicates that all delimited algorithms are suitable for use in the Target, Period, and Setting sets.

After testing with the ZeroR algorithm and identifying the baseline of the classification accuracy metric, tests with the OneR algorithm were performed. This algorithm generates rules based on a single predictor attribute of the applied dataset, that is, it creates rules for each dataset attribute and selects the rules with the lowest error rate as the only rules to use [36]. Table 4 presents the test performed by applying the OneR algorithm to the Target, Period, and Setting artificial datasets, and using the cross-validation model with the value 10.

For all sets, the algorithm defined the user attribute as having the lowest error rate among the other predictor attributes and the classify outputs were OutTargetNone, OutPeriodNone, and OutSettingNone respectively for the Target, Period, and Setting sets. However, the generated rules were not consistent with the context of notification management, as all sets were classified that notifications must not be notified to users. Thus, another test, presented in Table 5, was performed with the same configurations of the previous test and using preprocessing to remove the values OutTargetNone, OutPeriodNone, and OutSettingNone of the class attributes of each artificial dataset.

**Table 4.** OneR test.

| Artificial Dataset | Rules |
|---|---|
| Target | Member1 > OutTargetNone<br>Member2 > OutTargetNone<br>Member3 > OutTargetNone |
| Period | Member1 > OutPeriodNone<br>Member2 > OutPeriodNone<br>Member3 > OutPeriodNone |
| Setting | Member1 > OutSettingNone<br>Member2 > OutSettingNone<br>Member3 > OutSettingNone |

**Table 5.** OneR test with preprocessing.

| Artificial Dataset | Rules |
|---|---|
| Target | InMember1 > OutMember1<br>InMember2 > OutMember2<br>InMember3 > OutMember3<br>InAll > OutAll |
| Period | InMorning > OutMorning<br>InAfternoon > OutAfternoon<br>InNight > OutMorning<br>InDawn > OutMorning |
| Setting | Relevance1 > OutCurrent<br>Relevance2 > OutVibrate<br>Relevance3 > OutSilent |

Based on Table 5, we obtained rules consistent with the context of notification management compared to Table 4. In the Target dataset, the attribute selected with the lowest error rate inTarget was the most relevant, that is, the algorithm identified notifications that should be notified to certain users are actually notified to them. Therefore, in the Period dataset, the selected attribute was inPeriod, indicating that notifications that arrive in a given period in the environment must be notified in the same period. It just does not happen with the InNight and InDawn values. Finally, in the Setting

dataset, the selected attribute was activity, showing that the more relevant the activity the environment user is performing, the less disturbance there will be when a notification is notified.

Therefore, with the tests performed, it was identified that the artificial datasets generated are suitable for use over the management context in receiving notifications in smart environments. It was observed that the datasets have an adequate baseline of the classification accuracy metric and that the delimited algorithms were above it, and the previously generated rules followed a logic consistent with the proposed context of the work. Thus, we chose to use, in the remaining tests of the next subsections, the datasets without preprocessing, because the values OutTargetNone, OutPeriodNone, and OutSettingNone still have relevance in each dataset.

### 5.2. Adjustable Parameters Test

We also developed adjustable parameter tests between algorithms to identify which parameters are the best. Thus, for the J48 algorithm, we tested the parameter that determines the use of pruning in the decision tree. In the KNN algorithm, we evaluated the K parameters and similarity calculation. The kernel and cost parameters were tested on the SVM algorithm. Finally, for the MLP algorithm, we tested the parameters of the maximum amount of iterations, learning rate, momentum, and the number of neurons in the hidden layer. The PRISM and NB algorithms do not have adjustable parameters, because of this, no tests were performed with them. The tests were performed with the three artificial datasets Target, Period, and Setting by verifying the classification accuracy metric and using cross-validation with a value of 10. Table 6 presents the adjustable parameter test results for each algorithm.

**Table 6.** Test adjustable parameters.

| Algorithms | Parameters |
|---|---|
| J48 | Pruning: use |
| KNN | Distance calculation: Euclidean<br>K: 3 |
| SVM | Kernel: Linear<br>Cost: 10 |
| MLP | Iteration: 500<br>Learning rate: 0.3<br>Momentum: 0.2<br>Hidden layer neurons: attribute + class |

In Table 6, we identify the best adjustable parameters for each classification algorithm. Thus, these parameters were used in the experiments presented in the following subsections.

### 5.3. CPU Time Test

After defining the best adjustable parameters in each algorithm, we developed tests of the CPU time metric for predictor model training and the CPU time for the classification of new data instances. The CPU time metric was listed as relevant to the scope of the work because, in the context of IoT, there is a great need for information to be transmitted in real-time to both users and IoT devices. Therefore, the tests performed were conducted on each artificial dataset Target, Period, and Setting and using predictive model cross-validation with a value of 10. Tables 7 and 8 show the test results of training and classification time, respectively.

With the results presented in Table 7, the NB, KKN, and J48 algorithms obtained the shortest times, respectively, following the PRISM and SVM algorithms, and the MLP algorithm obtained the longest training time. At the end of the test, it was identified that the KNN algorithm is better and the MLP algorithm is the worst in the training time of a predictor model. Therefore, for Table 8, the algorithms

J48, PRISM, and NB obtained the shortest times, respectively, followed by the algorithms MLP, SVM, and KNN. At the end of the tests, it was identified that the J48 algorithm is the best and the KNN algorithm is the worst in the classification time of a new data instance.

**Table 7.** CPU time test for training.

| Artificial Dataset | NB | PRISM | J48 | KNN | SVM | MLP |
|---|---|---|---|---|---|---|
| **Target** | 0.47 ms | 12.03 ms | 2.19 ms | 0.47 ms | 43.44 ms | 15.815,31 ms |
| **Period** | 0.47 ms | 124.69 ms | 5.63 ms | 0.47 ms | 417.66 ms | 15.969,53 ms |
| **Setting** | 0.94 ms | 48.91 ms | 3.75 ms | 0.16 ms | 472.66 ms | 14.249,53 ms |
| **Average** | 0.62 ms | 61.87 ms | 3.85 ms | 0.36 ms | 311.25 ms | 62.011,45 ms |

**Table 8.** CPU time test for classification.

| Artificial Dataset | NB | PRISM | J48 | KNN | SVM | MLP |
|---|---|---|---|---|---|---|
| **Target** | 1.25 ms | 0.16 ms | 0.63 ms | 116.56 ms | 6.25 ms | 2.19 ms |
| **Period** | 1.56 ms | 1.87 ms | 0.31 ms | 123.59 ms | 47.03 ms | 3.13 ms |
| **Setting** | 1.72 ms | 0.47 ms | 0.31 ms | 123.28 ms | 49.06 ms | 1.25 ms |
| **Average** | 1.51 ms | 0.83 ms | 0.41 ms | 121.14 ms | 34.11 ms | 2.19 ms |

*5.4. Classification Precision Metric Test*

Regarding the comparison of the classification algorithms on the classification precision metric, the six classification algorithms were applied to the three artificial datasets. For each algorithm, we used the parameters delimited by Table 6 and the test with predictor model cross-validation equal to a value of 10. Therefore, the algorithms were executed 30 times in each artificial dataset. For each run, we used the random seed generator with different values following an increasing pattern, starting from the value 1. This was done to obtain greater randomness in the metric results, and for the Friedman test Section 5.5. Therefore, each result of the classification accuracy metric for each seed was placed in an Excel spreadsheet to calculate the average of the metric between the algorithms.

Unlike the other tests performed, the classification algorithms were executed in the NetBeans programming IDE using the WEKA package that contains the main features of the tool. This was necessary due to a large number of iterations performed in each algorithm, as it would take a long time if it were performed in GUI technology. Table 9 presents the average results of each randomization seed from the classification accuracy metric.

**Table 9.** Comparison of classification accuracy metric.

| Artificial Dataset | NB | PRISM | J48 | KNN | SVM | MLP |
|---|---|---|---|---|---|---|
| **Target** | 100% | 100% | 100% | 99.99% | 100% | 100% |
| **Period** | 80% | 96.87% | 97.49% | 97.22% | 87.53% | 99.96% |
| **Setting** | 85.26% | 98.86% | 99.54% | 99.89% | 86.89% | 100% |

From Table 9, we observed that all algorithms obtained the metric value above 70% in all artificial datasets. Analyzing the comparison of the algorithms separately in each dataset, in Target, almost all the algorithms obtained the percentage with the maximum value, only the KNN algorithm obtained a lower value. In the Period dataset, the MLP algorithm obtained the best result, the PRISM, J48, and KNN algorithms obtained similar values close to the MLP algorithm value, and the NB and SVM algorithms obtained lower values than the other algorithms. In Setting, the same behavior of the previous dataset was maintained, being the MLP algorithm with the best value, the PRISM, J48, and KNN algorithms with values similar to the MLP value, and finally the NB and SVM algorithms with lower values than other algorithms.

Upon considering the comparison made of the classification algorithms on the artificial datasets and analyzing it in general, it was observed that the MLP algorithm, in all sets, obtained the best

classification performance when compared to other algorithms, following the PRISM, J48, and KNN algorithms whose performances were very close to the MLP algorithm. Finally, the NB and SVM algorithms have always obtained lower-ranking performances than the other algorithms. Therefore, it was necessary to perform the Friedman statistical hypothesis test to verify if there is a statistical difference between the compared classification algorithms. This test determines whether one algorithm has better or worse rating performance than another, even if its rating accuracy metric percentage is higher or lower.

*5.5. Friedman Test*

As for Friedman's statistical hypothesis test, we proceeded from the comparison made in the previous subsection. The test was implemented in the R programming language, using the RStudioIDE together with the Excel program to transform quantitative data (classification accuracy) into ordinal qualitative data (ranking positions). As the Friedman test requires ordered data, the Excel program was used to generate a ranking for each artificial dataset. Each ranking has the positions of classification algorithms in each value of the seed generating randomness. Thus, the algorithm that obtains the highest value of the classification precision metric at a given seed value will be ordered first in the ranking and so on with the other algorithms according to their values. The classification precision metric values of each algorithm and seed were obtained through the 30 executions performed in the comparison of Section 5.4. The rankings have the positions of each algorithm in each seed value, as well as the average rankings for each algorithm. After creating the rankings of each artificial dataset, files were generated from them in CSV format with only the positions of the algorithms in each seed value, for import into IDE RStudio and thus perform the Friedman test.

For a better understanding of the positions of the algorithms in each artificial dataset, the data is presented Table 10, which reflects directly with the classification accuracy averages of Table 9. Thus, in all artificial datasets the MLP algorithm gets the first position, the J48, KNN, and PRISM algorithms alternating in second, third, and fourth position, and the SVM and NB algorithms in fifth and sixth position, respectively. This did not happen when there was a tie between the algorithms.

**Table 10.** Average ranking placements.

| Artificial Dataset | NB | PRISM | J48 | KNN | SVM | MLP |
|:---:|:---:|:---:|:---:|:---:|:---:|:---:|
| **Target** | 3.45 | 3.45 | 3.45 | 3.75 | 3.45 | 3.45 |
| **Period** | 6.0 | 3.85 | 2.15 | 3.0 | 5.0 | 1.0 |
| **Setting** | 6.0 | 4.0 | 3.0 | 2.0 | 5.0 | 1.0 |

Friedman's test was performed using an external package called "tools for R," using the CSV files generated from the ranking of each artificial dataset [37]. Figure 9 presents Friedman's tests performed with the Target, Period, and Setting datasets.

In the Target dataset, all algorithms are statistically equal and obtained their averages from similar positions, only the KNN algorithm obtained its slightly lower average position. Starting with the Period dataset, the MLP algorithm obtained the best position average and the NB algorithm the worst. The MLP, J48, and KNN algorithms were the most outstanding in this set, proving that they are statistically equal. This also happens for the Setting dataset, which has the same behavior as the previous set, with only the second and third place settings varying between the KNN and J48 algorithms.

Among all Friedman tests performed with artificial datasets, the algorithms that showed the most satisfactory results were the MLP, J48, and KNN algorithms since they were always among the first three ranking positions on the classification accuracy metric and proving that their classification performances are statistically equal in the context of the proposed work. The NB, SVM, and PRISM algorithms have always got the worst positions in all datasets, except when referring to the Target set.

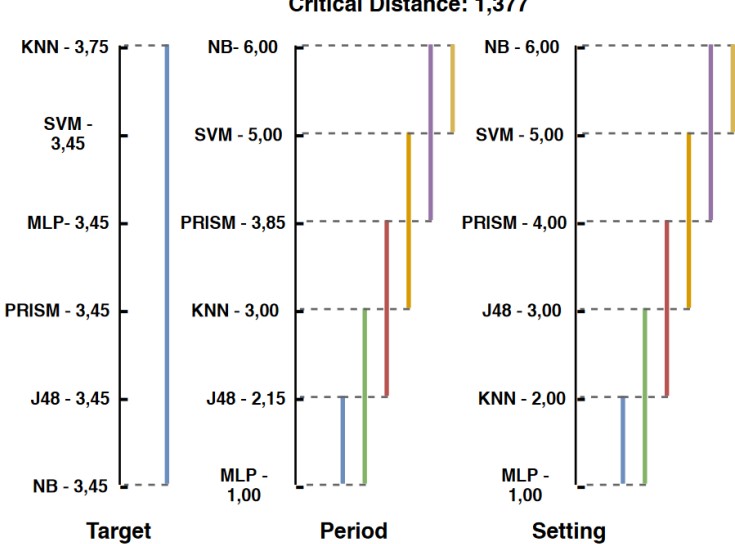

**Figure 9.** Friedman test.

With the experiments performed so far in this section, it was defined that the J48 algorithm is the most suitable to be used in the application scenario test. Taking into consideration the classification accuracy metric and the CPU time metrics analyzed, the algorithm has the best performance in both accurately classifying notifications for which user, time of day and type of device configuration, notification for the response time of classification, and training of the predictor model. Therefore, in the application scenario test, its behavior will be tested in a real scenario in the context of notification management in smart environments.

*5.6. Application Scenario Test*

To test the J48 classification algorithm implemented in the PRINM DM mechanism from the attributes and their values of Figures 4 and 5, the application scenario described in Section 4 was used. The predictive attributes used in the application scenario were the same as those used to generate the artificial Target, Period, and Setting datasets. Based on this, the J48 algorithm was trained with the three artificial datasets, thus, creating three decision trees that classify for which user, period, and device configuration notifications must be notified throughout the application scenario. The Target set tree has a size of 36 nodes, 27 of these are leaf nodes. The tree of the Period set has a size of 235 nodes, 169 of these are leaf nodes. Finally, the tree of the Setting set had a size of 224 nodes, 159 of these are leaf nodes.

Predictive data were collected from an artificial and a real source. For the artificial source, the data were collected from the users' actions in the application scenario for the values of the attributes user, profile, environment, and activity. For the real source, data were collected from the mobile *PRISER: Notification Collector* application developed by [38] for the values of the attributes status, inPeriod, and inTarget. Data from both sources were merged, thus, generating three unclassified datasets (test sets). Artificial source data are presented in Tables 11–13 for each user.

**Table 11.** Customer user data.

| Period | Profile | Environment | Activity |
|---|---|---|---|
| 10:10–10:25 ∣ Morning | Guest | Show Room ∣ Public | 1 |
| 10:25–11:50 ∣ Morning | Guest | Sector of Sales ∣ Private | 2 |
| 16:00–16:15 ∣ Afternoon | Guest | Show Room ∣ Public | 1 |
| 16:15–16:30 ∣ Afternoon | Guest | Sector of Sales ∣ Private | 2 |
| 16:30–18:00 ∣ Afternoon | Basic | Office ∣ Restrict | 3 |

**Table 12.** Employee user data.

| Period | Profile | Environment | Activity |
|---|---|---|---|
| 7:00–9:30 ∣ Morning | Basic | Show Room ∣ Public | 2 |
| 9:30–10:00 ∣ Morning | Advanced | Kitchen ∣ Private | 1 |
| 10:00–11:50 ∣ Morning | Basic | Show Room ∣ Public | 2 |
| 11:50–14:00 ∣ Afternoon | Advanced | Kitchen ∣ Private | 1 |
| 14:00–16:30 ∣ Afternoon | Basic | Show Room ∣ Public | 2 |
| 16:30–18:00 ∣ Afternoon | Basic | Sector of Sales ∣ Private | 3 |

**Table 13.** Owner user data.

| Period | Profile | Environment | Activity |
|---|---|---|---|
| 7:30–8:30 ∣ Morning | Administrator | Sector of Sales ∣ Private | 2 |
| 8:30–10:25 ∣ Morning | Administrator | Office ∣ Restrict | 3 |
| 10:25–11:50 ∣ Morning | Administrator | Sector of Sales ∣ Private | 2 |
| 11:50–14:00 ∣ Afternoon | Administrator | Kitchen ∣ Private | 1 |
| 14:00–16:30 ∣ Afternoon | Administrator | Sector of Sales ∣ Private | 2 |
| 16:30–18:00 ∣ Afternoon | Administrator | Office ∣ Restrict | 3 |
| 18:00–19:00 ∣ Night | Administrator | Kitchen ∣ Private | 1 |
| 19:00–20:00 ∣ Night | Administrator | Office ∣ Restrict | 2 |
| 20:00–22:00 ∣ Night | Administrator | Office ∣ Restrict | 1 |

Real source data was collected by the mobile application installed on three different mobile devices, representing the three users of the application scenario. The application collected notifications from the three mobile phones for 24 h, generating information in the JSON format of each notification. For each JSON, only the information corresponding to the attributes status, inPeriod, and inTarget were extracted, which coincide with the periods of each user within the dealership. Therefore, they were extracted for the client user from 10:10 a.m. until 11:50 a.m. and 04:00 p.m. until 06:00 p.m., for the employee user from 7:00 a.m. until 06:00 p.m., and the owner user from 7:30 a.m. until 10:00 p.m. Other notifications outside these times of each user was discarded. Therefore, 1377 notifications were received within the defined times for the client user (mobile phone 1), of these, 209 were extracted; 813 for the employee user (mobile phone 2), of these, 399 were extracted; and 413 for the owner user (mobile phone 3), of these, 333 were extracted.

The application scenario test aims to analyze the behavior and classification performance of the J48 algorithm on the classification accuracy metric using predictive data from the test sets and classifying them. However, after merging the predictive data from the two sources for generating test sets, they were also merged into them, specifically, data classes following the same logic as the script that generated the artificial dataset Target, Period, and Setting. This was necessary because it would not be possible to demonstrate all the values of the class attributes that the J48 algorithm would classify from the predictive data of the test sets in the application scenario. Therefore, the test sets merged with the client, employee, and owner user class data were applied to the decision trees created by the J48 algorithm. The results presented show that the J48 algorithm was able to classify with 100% accuracy new unclassified data inserted in each decision tree. Thus, it is concluded that the J48 algorithm has the proper behavior regarding the classification performance of notifications received in smart environments and that it is the most viable for implementation in the PRINM DM mechanism that will be developed in UBIPRI.

## 6. Conclusions

This paper presents a comparison of classification algorithms for managing receiving notifications in smart environments. UBIPRI was used as a base, which is an MW that has as its primary objective the treatment of privacy in smart environments. Therefore, an architecture was proposed that presents the performance of PRINM together with the PRIPRO and PRICRI modules provided by the addressed

MW. With it, it was possible to manage notifications that are received in environments that UBIPRI operates, obtaining data from the PRINM DM mechanism for which user, time of day, and device configuration.

The methods and materials were fundamental for understanding the complexity involved in developing the proposed solution of this work. PRINM is an implementation to be developed in UBIPRI to maintain the privacy of environments in the context of receiving notifications. Attributed from an intelligent DM mechanism that is implemented with a classification algorithm that receives attributes regarding the environment, users, and notifications. The manager ensures the delivery of notifications according to the privacy employed by UBIPRI in the smart environment in which it operates.

The activities carried out were the delimitation and use of NB, J48, KNN, MLP, PRISM, and SVM algorithms, delimitation and application of Friedman test, generation of artificial datasets, tests and comparisons of classification algorithms and scenario test of application. The delimited algorithms obtained high efficiency in the tests in which they were applied, satisfactorily contributing to the developed solution of the work. With Friedman's statistical hypothesis test, it was possible to identify statistical differences in the classification performance of the classification algorithms. Regarding the artificial datasets, they were suitable for use. The experiments identified the best tunable parameters, CPU time for training and classification, and classification accuracy metric values between classification algorithms. Finally, the application scenario test presented the applicability of PRINM on the notification management architecture.

Regarding the use of the Friedman test, it was of great importance for the development of the work solution. For without its application, it would not be possible to identify that the algorithms MLP, J48, and KNN have the same classification performances. It is thus proving among the three and from the other experiments, that the J48 algorithm is the most viable to implement in the PRINM DM mechanism that will be developed in UBIPRI. Regarding the CPU time test, besides the classification accuracy metric that was the main one addressed in the work, the CPU time metric was also pertinent for the analysis of the classification algorithms in the context of management in receiving notifications in smart environments. It is possible to measure the time of construction of a predictor model and classification of new data, since applications that use the IoT concept need information transferred and generated in real-time when requested.

The first item for future work is the development of PRIPRO and PRINM, and to assign it in UBIPRI, based on the coined architecture and the implementation of the J48 algorithm in the DM mechanism. It is also suggested to optimize artificial datasets by inserting new attributes, as well as using other MW modules addressed, so that notification management becomes more meticulous about the privacy that UBIPRI employs in smart environments. Thus, the following topics for work are proposed:

- Extend the comparison to use real data;
- Extend the comparison to include other classification algorithms;
- Extend the comparison to include other predictive attributes;
- Extend the comparison to include other classifying attributes;
- Extend the comparison to include other metrics and statistical tests;
- Develop the notification manager PRIPRO and PRINM;
- Perform tests on real application scenarios;

**Author Contributions:** Conceptualization, J.A.M.; Investigation, J.A.M., I.S.O. and L.A.S.; Methodology, J.A.M., I.S.O. and L.A.S.; Project Administration, L.A.S., V.R.Q.L.; Resources, V.R.Q.L., G.V.G. and J.D.P.S.; Supervision, V.R.Q.L., L.A.S. and J.D.P.S.; Validation, I.S.O., L.A.S. and V.R.Q.L.; Writing—original draft, J.A.M., L.A.S.; Writing—review and editing, I.S.O., L.A.S., A.S.M., G.V.G., J.D.P.S. and V.R.Q.L. All authors have read and agreed to the published version of the manuscript.

**Funding:** This research was supported in part by the Programa de Bolsas Universitárias de Santa Catarina/SC—UNIEDU for Instituições Comunitárias de Ensino Superior and CAPES. This study was financed in part

by the Coordenação de Aperfeiçoamento de Pessoal de Nível Superior—Brasil (CAPES)—Finance Code 001. Supported by project PLATAFORMA DE VEHÍCULOS DE TRANSPORTE DE MATERIALES Y SEGUIMIENTO AUTÓNOMO — TARGET. 463AC03, SA063G19. The research of André Filipe Sales Mendes has been co-financed by the European Social Fund and Junta de Castilla y León (Operational Programme 2014–2020 for Castilla y León, EDU/556/2019 BOCYL). Project co-financed with Junta Castilla y León, Consejería de Educación and FEDER funds, including a cooperation with the project international cooperation project Control and History Management Based on the Privacy of Ubiquitous Environments—Brazil/Portugal. This work has been supported by the Junta De Castilla Y León - CONSEJERIA DE ECONOMIA Y EMPLEO: Sistema para la simulación y entrenamiento en técnicas avanzadas para la prevención de riesgos laborales mediante el diseño de entornos de realidad híbrida with ref J118.

**Conflicts of Interest:** The authors declare no conflicts of interest.

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
