# Peer review of "PRIPRO: A Comparison of Classification Algorithms for Managing Receiving Notifications in Smart Environments"

_applsci, doi:10.3390/app10020502_

Round 1

Reviewer 1 Report

The authors included additional sections to improve the previously reviewed manuscript. These additions provided to some extent some clarity. However, these additions added more confusing because of the poor quality of presentation in some instances and English/grammar issues.

It is not clear why section 5 "Related Works" was included near the end of the manuscript. This section included information that is more relevant to introduction and discussion. Reorganization of this section is recommended.

Organize the manuscript to clearly show subheading of data, methods, results, and discussion sections. 

The authors should acknowledge that they use artificial dataset which far from reality and that the developed approach needs further testing.

The authors should acknowledge that they use artificial dataset which far from reality and that the developed approach needs further testing. Another option is to split the artificial data into two parts, one for developing and testing and the other subset for evaluating. 

Author Response

First and foremost, we would like to thank the reviewers for the contribution, which has proven to be very valuable in improving our paper. His comments have alerted us to several limitations that we were previously unaware of and that we have since attempted to address.

Reviewer 2 Report

The related work should come after the introduction

The machine learning theoretical part should be reduced and more emphasis should be on the actual work did by the researchers.

Conclusions should be extended.

Author Response

(The authors gave the same response as above.)

Round 2

Reviewer 1 Report

Reviewers’ comments

General

The revised version of the manuscript is an improvement compared with the previous version. However, the English is still a big issue that needs extensive editing. This weakens the manuscript and hinders the reader from understanding the presented research.

Here are some comments to help improve the manuscript

Line 68:  this sentence is not clear. Please revise for clarity.

Line 70:  'it' does it refer to this manuscript or to UBIPRI. Please clarify

Line 79: related “works” please use work instead. Do not use “works”. Please revise the entire manuscript for this issue

Line 80: First not “firstly”. The reader would expect to see this followed by second or an enumerate. No need to start the sentence with an enumerate just remote it.

Line 83: change “data set” to “dataset”. Revise the entire manuscript for this

Line 138-139: please delete this sentence. No need to repeat what this section provides. This was already mentioned in the introduction.

Line 185: “Classification task supervised learning”. This is not clear. classification task …what … may be provided ... or using ....?

Line 187-188: “However, There is a wide diversity of algorithms that exist within this task is  large, so it is necessary determine”. Please revise this sentence.

Line 251: we “listed”

Line 400: Figure 8 – provide a different color for each method NB, SVM, etcs.

Author Response

Dear Reviewer,

First, we would like to thank you for all the criticisms and suggestions that undoubtedly made our work more quality.

We changed the article to your requests, as you can see in the letter on PDF.

Reviewer 2 Report

The review version responded to all my previous comments.

Author Response

Dear Reviewer,

Thanks for your previous suggestions and we wish you a happy holiday!